# Optical Axons for Electro-Optical Neural Networks

**DOI:** 10.3390/s20216119

**Published:** 2020-10-27

**Authors:** Mircea Hulea, Zabih Ghassemlooy, Sujan Rajbhandari, Othman Isam Younus, Alexandru Barleanu

**Affiliations:** 1Faculty of Automatic Control and Computer Engineering at Gheorghe Asachi Technical University of Iasi, 700050 Iasi, Romania; alexb@tuiasi.ro; 2Optical Communications Research Group, Faculty of Engineering and Environment at Northumbria University, Newcastle upon Tyne NE7 7XA, UK; z.ghassemlooy@northumbria.ac.uk (Z.G.); othman.younus@northumbria.ac.uk (O.I.Y.); 3Huawei Technologies Sweden AB, 412 50 Gothenburg, Sweden; sujan@ieee.org

**Keywords:** optical neural networks, optical axons, optical signal fading, VLC

## Abstract

Recently, neuromorphic sensors, which convert analogue signals to spiking frequencies, have been reported for neurorobotics. In bio-inspired systems these sensors are connected to the main neural unit to perform post-processing of the sensor data. The performance of spiking neural networks has been improved using optical synapses, which offer parallel communications between the distanced neural areas but are sensitive to the intensity variations of the optical signal. For systems with several neuromorphic sensors, which are connected optically to the main unit, the use of optical synapses is not an advantage. To address this, in this paper we propose and experimentally verify optical axons with synapses activated optically using digital signals. The synaptic weights are encoded by the energy of the stimuli, which are then optically transmitted independently. We show that the optical intensity fluctuations and link’s misalignment result in delay in activation of the synapses. For the proposed optical axon, we have demonstrated line of sight transmission over a maximum link length of 190 cm with a delay of 8 μs. Furthermore, we show the axon delay as a function of the illuminance using a fitted model for which the root mean square error (RMS) similarity is 0.95.

## 1. Introduction

In recent years, several algorithms for machine learning-based stream learning have been reported in the literature. However, in dynamic and evolving environments, most reported models require retraining of the learned algorithms, which is not desirable. Artificial neural networks (ANNs), which mimic the human brain’s process of acquisition and processing of sensory information, have received a great deal of attention for a range of applications. Within this context, spiking neural networks (SNNs) have emerged as the most successful approach to model the behavior and learning features of biological neural networks (NNs), and to represent and integrate information in time, space, frequency, and phase domains. The SNN benefits from the increased computation power and accuracy compared with other types of NNs, including the traditional ANNs [1,2]. This is because an SNN operates based on discrete events (i.e., precise timing spikes), which make it sensitive to time-varying functions and random occurrence of events [3]. The SNN is potentially best suited for hardware implementation [4,5,6] because it is based on parallel operation using a significant number of high complexity neurons with the simple “integrate-and-fire” feature. In addition to the benefit of higher biological plausibility [7], a hardware-based SNN offers very low energy consumption [8,9,10] and efficient decoding of the data generated by event-based sensors [3].

Typically, spiking neurons (SNs) model the key characteristics of the neural cells, which are used for information processing and adaptability. These characteristics are (i) the spiking behavior, integration of the incoming stimuli, detection of the activation threshold, refractory period, and generation of excitatory or inhibitory stimuli; and (ii) properties related to the synaptic plasticity that include the long-term potentiation and depression, which increases or decreases the synaptic weights *w_s_* [11]. Typically, the communication between SNs is performed using the discrete stimuli, in which the intensity of the postsynaptic response depends on *w_s_* [11]. Implementing the SNs in the analogue hardware domain enables parallel transmission of multiple colors (i.e., wavelength division multiplexing (WDM)) between neurons [12]. Optical neural networks (ONN) have been investigated offering a number of features including (i) the computational power of the neuromorphic hardware; (ii) high processing speed and energy efficiency [13,14,15,16]; (iii) use of the license-free optical spectrum (i.e., ultraviolet, visible, and infrared bands); and (iv) lower crosstalk and immunity to radio-frequency induced electromagnetic interference [17].

The photonic technology was first introduced in hybrid feed-forward NNs [18,19,20]. Recently, all-optical neurons with sigmoid or Softplus activation functions were reported in [21,22], whereas in [23] the backpropagation algorithm was used to adjust the weights of all-optical synapses. In addition, significant research activities on the implementation of optical communications between different types of optical spiking neurons were reported in [24,25]. Following the first demonstration of the integration of a small all-optical NN in silicon [26], more studies of implementations of all-optical synapses [27,28,29] have been published. Because adaptability is an important feature of NNs, the main goal of the design of optical synapses is to model the biological learning rules. Such adaptable synapses are based on electro-absorption modulators [30], memristors [31], and photoconductive materials, such as graphene with carbon nanotubes [32], amorphous oxide semiconductors [33], and carbon nanotube field-effect transistors [34]. In [35], the adaptive optical synapses were implemented using an organic photosensitive material that stimulates contractile artificial muscles in response to light intensity.

In NNs, the role of synapses is to transmit weighted signals to the postsynaptic neurons (posts). In early models of electro-optical neurons (EON), which were used in computational NNs, *w_s_* were transmitted using light intensity modulation (LIM) [18,19,20]. This concept was adopted later in significantly faster photonic neural networks (PNNs) using the silicon photonic weight banks [13], [36]. In advanced PNNs, the photonic spike processing is performed using ultrafast laser neurons [16,37], in which adjustable *w_s_* are implemented using different types of tunable silicon micro-ring resonators [38]. Note, in LIM-based ONNs the optical signal level is substantially reduced following transmission over several nonlinear layers, which can be mitigated by means of optical amplification [39]. Similarly, in free space-based ONNs, the weighted signal is affected by the light intensity fluctuations during propagation, which depends on the transmission span. One possible option to mitigate the effect of intensity variation is to use electro-optical synapses, in which *w_s_* is encoded within the optical pulse width [12]. However, the stray capacitance of transistors affects transmission of lower values *w_s_*.

In this work, we introduce a novel concept of an optical axon (OA) for implementation of electro-optical SNNs, which uses optical wireless transmission between neural areas. The proposed optical axon overcomes the disadvantage of transmitting the weighted optical signals [39] (i.e., reduced implementation complexity of online learning mechanisms) and improves the ONN’s tolerance to the variation of light intensity. The OA can be used in optical connections to the upper neural layers of mobile neuromorphic sensors for conversion of analogue inputs into the spiking frequency [40]. Analog-to-spike conversion has been used in the implementation of neuromorphic sensors in artificial olfactory systems [41,42,43] and in tactile sensors based on memristors [44]. Other neuromorphic sensors have been used for (i) control of the contraction force of shape memory alloy actuators [45]; and (ii) rotation control of robotic junctions [46].

To validate the OA, we used an ONN, in which both *w_s_* and plasticity are implemented in the analogue hardware domain, and visible light communication (VLC) for axonal transmission. For quantifying the impact of the free space channel on ONN, we propose a novel model for estimation of the axon’s delay based on the amount of light being converted into an electrical signal. It should be noted that: (i) the work is focused on increasing the communication range between the neural areas and not on the implementation of ultrafast NNs; and (ii) analogue neurons, which operate in the kHz range, are a cost-effective option.

The remainder of the paper is organized as follows: Section 2 details the schematic of the OA and the EON modules. Section 3 and Section 4 present the experimental setup for validating the OA and the obtained results. The influence of the axon delay on the SNN’s activity is evaluated in Section 5, which is followed by a discussion section related to the possible use of optical axons in significantly faster SNNs. The paper ends with the conclusions.

## 2. Electro-Optical Spiking Neuron

Figure 1 shows the basic structure of the EON implemented in analogue hardware, which includes an electronic soma (SOMA) for information processing, and at least one electronic synapse (SYN), which are used for weight storage and adaptability. The SOMA and SYN are connected optically by an OA for transmitting the trigger signals for activating the synapses. The electronic synapse, which transmits the weighted signals, is hardwired to the SOMA in order to reduce the effect of the optical induced noise sources. The SOMA and SYN represent the main elements of a bioinspired neuron model, which were implemented in analogue hardware reported in our previous work [47,48] and is adopted in this work.

A number of different forms of neuromorphic hardware with bioinspired plasticity rules have been reported in the literature [49], including the neuron model used in this work [47]. The SOMA models the integration of input stimuli detection of the activation threshold and refractory period as in [12], and the spike generation and the plasticity mechanisms are modeled by the SYN [47]. Note that in our previous work [12] the optical channel was implemented between the SYN output and the postsynaptic SOMA input, whereas in the current research the optical connection is between the SOMA output and the SYN input.

### 2.1. The Analogue Modules of the EON

Figure 2 presents the schematic circuit diagram of the SOMA and SYN in which the axon is hardwired as in the original neuron design [47,48]. According to Figure 2a, during the neuron idle state, the voltage level $V_{\mathrm{IN}}$ across the capacitor $C_{M}$ is set to $V_{\mathrm{EQU}}$, which represents the equilibrium voltage of the artificial neuron.

On activation of the SOMA, $V_{\mathrm{IN}}$ reaches the neuron activation threshold voltage $V_{\mathrm{TH}}$, which for this neuron model is the transistor $T_{1}$ base-emitter voltage $V_{\mathrm{BE}}$. During activation, the SOMA generates an active low pulse with a fixed width at $S_{\mathrm{OUT}}$ for triggering, i.e., the SYNs which are connected to its output. The pulse width is determined by the capacity $C_{R}$ which is fully discharged during the neuron idle state. The discharge of $C_{R}$ determines the saturation time of $T_{1}$ following the time interval $\Delta t_{N}$ for all neuron activations. Note, the initial charge of $C_{R}$ does not change with $f_{N}$, therefore the saturation time of $T_{1}$ and consequently the activation of OA are delayed by $\Delta t_{N}$.

The input voltage across $R_{E}$ is given by:(1)$$V_{\mathrm{IN}}=V_{0}+\left(V_{\mathrm{charge}}-V_{0}\right)\left(1-e^{-\frac{t}{\tau_{C}}}\right)$$
where $V_{0}$ is the initial voltage across $C_{M}$, $\tau_{C}=R_{E}C_{M}$, $V_{\mathrm{charge}}=(V_{E}^{\mathrm{PRE}}-V_{\mathrm{EQU}})R_{M}/(R_{E}+R_{M})$, and $V_{E}^{\mathrm{PRE}}$ is the amplitude of the excitatory spike. The inhibitory stimulus (for which amplitude is typically $V_{I}^{\mathrm{PRE}}<V_{\mathrm{EQU}}$) is described by:(2)$$V_{\mathrm{IN}}=V_{\mathrm{discharge}}+(V_{0}-V_{\mathrm{discharge}})e^{-\frac{t}{\tau_{D}}}$$
where $\tau_{D}=R_{I}C_{M}$, $R_{I}$ is the emitter resistor of $T_{5}$, and $V_{\mathrm{discharge}}=\left(V_{\mathrm{EQU}}-V_{I}^{\mathrm{PRE}}\right)R_{I}/\left(R_{I}+R_{M}\right)$ is the discharge from $C_{M}$. Note, if the SOMA is not activated, the equilibrium value of $V_{\mathrm{IN}}$ is restored via $R_{M}$.

Considering that the analogue modules of the EON use transistors with β=400, the model is then simplified for the case in which transistors are in the saturation or cut-off modes. During SOMA activation and when $T_{1}$ is in the saturation mode, $C_{R}$ is charged until $T_{1}$ is blocked. Thus, $C_{R}$, $R_{\mathrm{RU}}$, and $R_{\mathrm{RD}}$ determine the duration of the SOMA activation. At the same time, $C_{M}$ is discharged through $R_{F}$ to the forward voltage $V_{\mathrm{SCH}}<V_{\mathrm{BE}}$ of the Schottky diode. This mimics the behaviour of the neural cell input potential, which is below the equilibrium voltage following activation. When the SOMA is activated, $V_{U}$ is pulled to $V_{U0}$, which varies with $V_{L}$ (i.e., the voltage stored in the learning capacitor $C_{L}$), as shown in Figure 2b.

Note that $V_{L}$, which represents the synaptic weight, determines the saturation period of $T_{4}$ and consequently the generated spike duration and its energy. This excitatory output is given by:(3)$$V_{E}=\left\{\begin{array}{c} V_{\mathrm{SPK}},\mathrm{if}\;V_{U}<V_{\mathrm{DD}}-V_{\mathrm{EB}}\;\\ V_{\mathrm{IN}}^{\mathrm{POST}},\;\mathrm{otherwise} \end{array}\right.$$
where $V_{\mathrm{SPK}}$ = $V_{\mathrm{DD}}$ for typical operation frequencies in the range of kHz, $V_{\mathrm{IN}}^{\mathrm{POST}}$ is the input potential of the post connected to $N_{\mathrm{OUT}}$, and $V_{\mathrm{EB}}$ is the emitter-base of $T_{4}$. Note, with reference to Figure 2b the capacitor $C_{F}$ is discharged completly via $R_{F}$ during neuron activations thus maintaining the spike’s amplitude. However, using a set of $R_{F}$ and $C_{F}$, the spike’s amplitude can be reduced due to the high activation frequency thus leading to reduced frequency of the postsynaptic neurons. From the biological point of view, this behavior of the electronic neuron can model the spike-frequency adaptation of the natural neurons, which shows a decrease in the spiking frequency of neurons following long activation periods [50,51].

The excitatory spike is converted into an inhibitory stimulus by an additional transistor $T_{5}$ connected as in Figure 2b. Considering that the response time of $T_{5}$ is negligible, the inhibitory output is given as:(4)$$V_{I}=\left\{\begin{array}{c} 0,\mathrm{if}\;V_{U}<V_{\mathrm{DD}}-V_{\mathrm{EB}}\;\\ V_{\mathrm{IN}}^{\mathrm{POST}},\;\mathrm{otherwise} \end{array}\right.$$

Each SYN generates a spike at their output i.e., $N_{\mathrm{OUT}}$, which can be excitatory or inhibitory depending on the position of the switch S. Note that, from the biological point of view, a neuron can include only excitatory or inhibitory synapses and not both. Thus, *S* is in the same position for all synapses in the AN. Taking into account that a neuron includes at least one synapse, $T_{3}$ included in the first SYN, see Figure 2b, is used instead of $T_{2}$ as shown in Figure 2a.

For this neuron model, increasing *w_S_* implies increasing/deceasing $V_{L}$ with decreasing/increasing *w_S_* (i.e., increasing the energy of the generated pulse). During neuron activation, $V_{L}$ decreases via $R_{LU}+R_{LD}$, thus modeling the post-tetanic potentiation of the biological synapses. This presynaptic element of the synaptic plasticity represents weight adjustment when the neural cells are activated [52]. Furthermore, $V_{L}$ decreases when the postsynaptic activity occurs shortly after the activation of the SYN in concordance with the biological long-term potentiation [53]. This mechanism is implemented using $C_{A}$, which temporarily discharges $C_{L}$ by ΔQ. During the idle state that follows the neuron activation, the charge *ΔQ* is transferred back to $C_{L}$ if the post remains inactive. When the post is activated, the remaining ΔQ stored in $C_{A}$ is discharged through the input $S_{\mathrm{LTP}}$, which is connected to $S_{\mathrm{OUT}}$ of the postsynaptic SOMA. Note, the circuit AUX in Figure 2b is used to keep $T_{6}$ open until ΔQ is discharged.

### 2.2. The Proposed OA

The concept introduced in this paper represents the OA, which is adapted to implement communication between distanced neural areas and to increase ONN operation tolerance to the beam intensity fluctuations. The OA structure shown in Figure 3 replaces the hardwired axon, which connects the SOMA to SYNs. At the transmitter (Tx), the OA includes a LED panel, which is intensity-modulated by the signals from the SOMA via the p-channel MOSFET transistors and the load resistor as in Figure 4a. At the receiver (Rx), an optical Rx (ORx) is used to regenerate the electrical signal, which is amplified and passed through a timer module for use by the SYN.

Note, the synaptic weight determines the generated pulse width *T_s_*, therefore the duration $T_{W}$ of the SYN activation signal is critical. To reduce the changes (time jitter) in $T_{W}$ due to the channel induced perturbations, the electronic synapse is activated by a timer NE555; see Figure 4b.

## 3. Experimental Setup

We developed an experimental testbed as shown in Figure 5 for evaluating the performance of the proposed system, in particular, the OA’s tolerance to the optical transmission range and the link misalignment. The system, with the key parameters given in Table 1, is composed of two OAs (i.e., OA_BLUE_ and OA_RED_) for connecting two somas, SOMA_B_ and SOMA_R_, to a synapse SYN, timer, SNN, optical filter, and ORx (Thorlabs PDA100A2).

Note, two SOMAs are used for activating the OA’s transmission wavelengths, OR’s sensitivity, and the drive current and illuminance for the LEDs. The SNN comprises SYNs and SOMAs, which are controlled by a microcontroller (μC). Experiments were performed in an indoor environment with an artificial room illuminance level of between 440 and 462 lux; see Table 1. At the receiving end, we used an optical filter to reduce the ambient light and therefore improve the signal-to-noise ratio. In [12,38], WDM systems with filters for parallel transmissions between optical neurons were reported, whereas a less costly version of WDM without filters was recently introduced in [54].

The output of the ORx is further amplified prior to being converted into a positive pulse with a fixed *T*_*w*_ using NE555; Figure 4b.

## 4. Results

The goal of the experiments was to evaluate the influence of the link length $L_{\mathrm{OA}}$ on the OA operation and how the system operates considering the link misalignment. The operation of the OA was analyzed independently for the red and blue LEDs using the same Rx. Because the spikes are transmitted using laser light sources in recent ONNs [13,36,37] due to speed, in this research we used LEDs due to the larger field of view which increases the tolerance to the link’s misalignment. The presented results are valid for OAs if activated simultaneously because the optical filters are used for channel isolation at the Rx. The two SOMAs are activated by a μC at sufficiently long periods when the SOMA’s input reaches the equilibrium voltage.

### 4.1. Delay Induced by the OA

Because spike timing is the critical parameter in the SNN and SNs are temporal coincidence detectors, we measured the variation of the axon delay considering the link length and misalignment. Figure 6a shows the recorded signals when SOMA_R_ activates the synapse SO through the OA, which generates a 47.4 μs long pulse delayed by dT. Each value of dT was obtained for five consecutive measurements performed using the persistence function. Based on the data measured using the oscilloscope, the maximum jitter of the axon delay due to the optical channel is less than 0.4 μs for the maximum link span of 190 cm. The axon delay is determined by the beam intensity, which depends on the optical link length and the misalignment angle between the Tx and the Rx. For the evaluation tolerance of the OA on the link misalignment, the Tx was rotated at steps of 10° while the Rx was kept fixed.

Figure 6b shows the variation of OA delay as a function of the rotation angle $\alpha_{\mathrm{OA}}$ for an optical link length $L_{\mathrm{OA}}$ of 50 cm. Note, due to the symmetric beam profile, the Tx is rotated in only one direction. The maximum misalignment angle is $\alpha_{\mathrm{OA}}^{}=60{}^{\circ}$, which is lower than half of the LED’s viewing angle $\alpha_{\mathrm{view}}$ of 125°. This implies that at least three LEDs should be used to provide 360° coverage. As shown, the OA delay increases exponentially with $\alpha_{\mathrm{OA}}$, with the blue LED displaying higher delays. Next, we investigated the delay dT as a function of the optical link span $L_{\mathrm{OA}}$ for both LEDs for $\alpha_{\mathrm{OA}}$ of 0° and 60°, as shown in Figure 7a. For $\alpha_{\mathrm{OA}}$ of 0° and maximum $L_{\mathrm{OA}}$, the measured dTs are 8 and 4.2 μs for the blue and red LEDs, respectively.

Note that the predicted axon delay $dT_{\mathrm{predicted}}^{L}$ for $\alpha_{\mathrm{OA}}=60{}^{\circ}$ and variable $L_{\mathrm{OA}}$ was determined using the measured beam profiles for $L_{\mathrm{OA}}$ and $\alpha_{\mathrm{OA}}$, which is given as:(5)$$dT_{\mathrm{predicted}}^{L}=\frac{dT_{\mathrm{measured}}^{\alpha_{OA}=60{}^{\circ}}}{dT_{\mathrm{measured}}^{\alpha_{OA}=0{}^{\circ}}}dT_{\mathrm{measured}}^{L}$$
where $dT_{\mathrm{measured}}^{L}$ is the delay for $\alpha_{\mathrm{OA}}=0{}^{\circ}$ and variable $L_{\mathrm{OA}}$. For the line of sight link the maximum measured delay $dT_{\max}=8\;\mu s$, thus the axon may not be operational for $L_{\mathrm{OA}}$ > 170 cm and $\alpha_{\mathrm{OA}}=60{}^{\circ}$, where the predicted delay is above $dT_{\max}$; see Figure 7a. $dT_{\max}$ for the red channel is significantly lower than that of the blue channel. This is due to the higher irradiance level of the red LED and higher responsivity of the ORx at red wavelength; see Table 1.

### 4.2. Model for the Axon Delay

The axon delay was estimated by developing a model based on illuminance $E_{V}$ that fits dT. Thus, knowing $E_{V}$ the photocurrent in terms of the photopic efficacy $V_{\lambda }$ and the Rx’s responsivity, ῃ can be determined as:(6)$$I_{PD}=\frac{E_{V}}{683\;V_{\lambda }}ῃ$$

In addition, the measured dT has a component denoted by $dT_{\mathrm{bias}}$, which does not depend on $E_{V}$. This delay is given by the response time of the electronics used between OR and SYN; see Figure 4b.

Because the Rx response depends on $I_{PD}$, we have the following functions as shown in Figure 7b:(7)$$f=\beta +\frac{\alpha }{dT_{\mathrm{optical}}}\mathrm{and}\;g=\ln\left(I_{\mathrm{PD}}\right),$$
where $dT_{\mathrm{optical}}=dT-dT_{\mathrm{bias}}$ and the parameters α and *β* are used for scaling $1/dT_{\mathrm{optical}}$ on $\ln\left(I_{PD}\right)$, which are given as:(8)$$\alpha =\frac{\left(g_{\max}-g_{\min}\right)}{\left[{\left(1/{\mathrm{dT}}_{\mathrm{optical}}\right)}_{\max\;}–{\left(1/{\mathrm{dT}}_{\mathrm{optical}}\right)}_{\min}\right]}\;and\;\beta =\mathrm{mean}\;\left(\alpha /{\mathrm{dT}}_{\mathrm{optical}}\right)-\mathrm{mean}\left(g\right)$$

The illuminance $E_{V}$ was determined for $L_{\mathrm{OA}}=5,\;10,\;30,\ldots ,\;190\;\mathrm{cm}$ using $E_{V}^{\mathrm{init}}$, which was measured for $L_{\mathrm{OA}}=10\;\mathrm{cm}$; see Table 1. Note the high similarity between f and g, for which the root mean square error (RMS) values are 0.98 and 0.95 for red and blue LEDs, respectively. Thus, the delay introduced by the OA is given by:(9)$$dT=\frac{\alpha }{\ln\left(Eῃ\right)-\beta }+dT_{\mathrm{bias}},$$
where $\alpha_{R}=4.8$ and $\beta_{R}=6.58$ for the red channel, and $\alpha_{B}=6.4$ and $\beta_{B}=6.65$ for the blue channel. The delay that yields the best RMS for both wavelengths is $dT_{\mathrm{bias}}=0.5\;\mu s.$ This value is practically plausible considering the parameters of OR, LM111, NE555, and BC848 that are used to activate the synapse.

## 5. Axon Delay Influence on Neuron Activity

In this section, we determined the tolerance of the SNN to the maximum axon delay. Typically, wireless communications are suitable for neural areas that are connected via OAs. To determine whether the delays affect SNN behavior, we implemented a network that includes two electronic synapses (SYN_1_ and SYN_2_), which stimulate a SOMA; see Figure 8a.

To simplify the test process and to ensure repeatability of the results, the synapses were activated by pulse trains with different delays dT generated using a microcontroller (see Figure 8b), which simulates the axon’s output. The activity of the postsynaptic neuron was evaluated by monitoring the SOMA activation period $T_{S}$ for dT in the range of $\left(0,\;8.4\;\mu s\right)$ with a resolution of 0.7 μs. The measurements show that, for the maximum dT, $T_{S}$ increases by less than $dT_{S}=8\;\mu s$. The natural neurons operate typically at a few tens of Hz [55] with the maximum frequency $f_{\mathrm{BRAIN}}=453\;\mathrm{Hz}$ observed in the human brain [56]. This gives the minimum time interval between neuron activations of $T_{\mathrm{BRAIN}}=2.2\;\mathrm{ms}$. Note, $dT_{S}$ is less than 0.5% of the inter-spike period $T_{\mathrm{BRAIN}}$, implying that the influence of the axon delay on the biologically plausible frequency of spiking neurons is negligible.

## 6. Discussions

The results show that light intensity fluctuations do not influence the duration of the SOMA activation, thus lessening the effects on transmission of synaptic weights. During the experiments, the light intensity varied significantly (i.e., by two orders of magnitude), which implies that implementation of LIM is not feasible because it directly converts the synaptic weights into light intensity [13,36]. Note that an ONN based on spiking neurons typically converts the synaptic weights into the spikes’ amplitude and duration [12], therefore conversion of spikes into optical pulses will be susceptible to light intensity fluctuations. This is because changes in the amplitude of the optical pulse means modified synaptic weights.

Conversely, the OA induces a delay, which depends on the optical channel characteristics (i.e., length, alignment, and components) that affect the SNN operation by changing the frequency of the spiking neurons, thus affecting the polychronization of the SNN [57]. Recent research shows that the precision of electronic neurons can be as low as 50 ns in the detection of temporal stimuli [58]. This order of magnitude for OA delay can be reached using faster electronics. As an example, dT can be reduced below 30 ns for OPL = 1 m (tp=3.3 ns) using the ultrafast Tx with the response time $t_{r}<6\;\mathrm{ns}$ [59], in addition to a fast Rx implemented with FPD310 ($t_{r}<0.5\;\mathrm{ns})$, comparator LMH7220 ($t_{r}<7\;\mathrm{ns})$, and fixed-width pulse generator based on ultra-fast logic gates ($t_{r}<9\;\mathrm{ns}$); see the OA structure in Figure 1. However, because increasing the communication distance between neural areas results in non-compensable delays, one possible option to reduce the OA influence on the SNN would be to decrease the neuron’s frequency. Low-speed analogue hardware platforms will increase the tolerance of the SNN to the increased transmission spans. The possible applications that can benefit from the proposed optical axons are bio-inspired systems, including neuromorphic sensors [4,40], with a built-in preprocessing neural layer. The tolerance of the OA to the variation of the channel length and link misalignment makes the OA suitable for exploration applications in which neuromorphic sensors are mobile. In addition, the OA could be used in the implementation of brain–computer interfaces [60] or in the recently tested brain-to-brain communications [61,62].

In practical applications, the maximum number of inputs $n_{\mathrm{IN}}$ and outputs $n_{\mathrm{OUT}}$ of the system will determine the number of parallel free space channels (i.e., free space with WDM) that could be used. For the single-color LED-based Txs, $n_{\mathrm{OUT}}$ is determined by the number of available wavelengths within the range of 250–1650 nm. Note, higher $n_{\mathrm{IN}}$ and $n_{\mathrm{OUT}}$ (i.e., up to 70) can be employed by using a combination of broad-band LEDs and narrow-band optical bandpass filters (OBPF), and a high-bandwidth PD. However, because of the limited dynamic range of the optical Tx and Rx and their dependency on λ, in addition to system complexity, about 35 channels should be used.

## 7. Conclusions

A common method to encode synaptic weights in optical neural networks is light intensity modulation. In free space-based ONNs, the weights are affected by beam fading due to variation in the distance between neural areas. This paper presented the theoretical and practical implementation of a new and efficient method to significantly reduce the effects of beam fading in free space optical-based neural networks. The proposed method is based on optical axons, which eliminate the requirement for synaptic weight transmission over the free optical channels. OAs could be used to implement the optical connections in multi-modular bioinspired systems, which include mobile neuromorphic sensors. For testing this novel concept, we determined the delay induced by two OAs with different parameters. Results showed that the delay is significantly influenced by the axon parameters of light intensity and the responsivity of the optical Rx. Thus, based on experimental data we developed a model to estimate the delay as a function of illuminance. The maximum axon length and the delay achieved were 190 cm and 8 μs, respectively, for the line of sight transmission. It was shown that the influence of the axon delay on the biologically plausible frequency of spiking neurons is insignificant for the considered SNN that operates in the kHz range. For faster SNNs, lower values of OA delay can be achieved using high-speed electronic circuitry. However, OA integration in ultrafast photonic NNs has limitations determined by the speed of light.

In future work we will (i) carry out testing and measurement of the associative learning mechanisms with optical axons used in ONNs, and increase the number of channels; and (ii) focus on the evaluation of an anthropomorphic robotic arm with the neuromorphic rotation and force sensors connected optically to the main SNN.

## Figures and Tables

**Figure 1 sensors-20-06119-f001:**
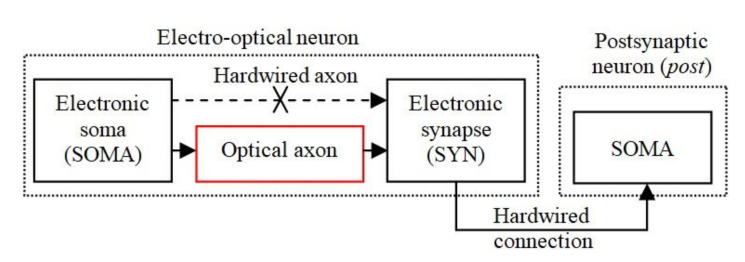
The general structure of the electro-optical neuron (EON) that includes a SOMA connected through the optical axon (OA) to a SYN.

**Figure 2 sensors-20-06119-f002:**
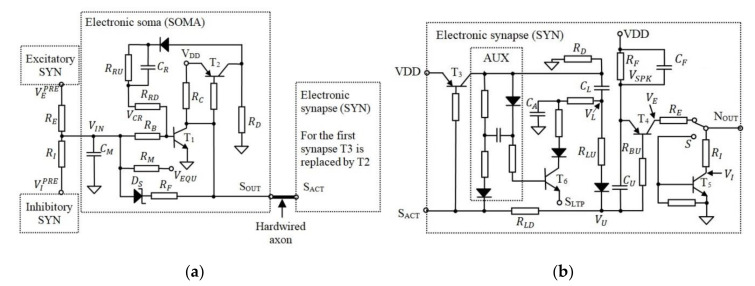
The schematic of (**a**) the electronic SOMA and (**b**) the electronic synapse.

**Figure 3 sensors-20-06119-f003:**
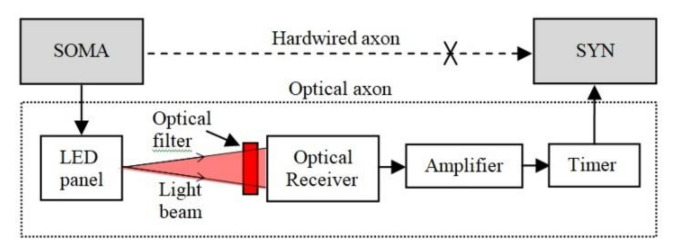
The general structure of the OA including the LED panel at the Tx and the optical Rx, amplifier and timer on the Rx side.

**Figure 4 sensors-20-06119-f004:**
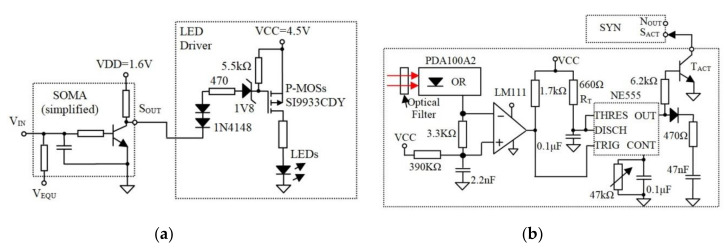
(**a**) The SOMA drives the LEDs using p-channel MOSFET transistors and (**b**) the Rx including the optical Rx and the additional circuitry that generate fixed-width pulses for SYN.

**Figure 5 sensors-20-06119-f005:**
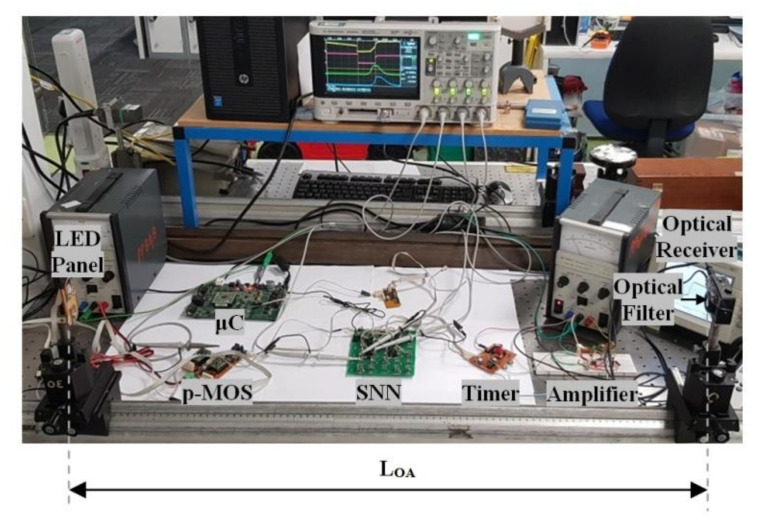
The experimental setup.

**Figure 6 sensors-20-06119-f006:**
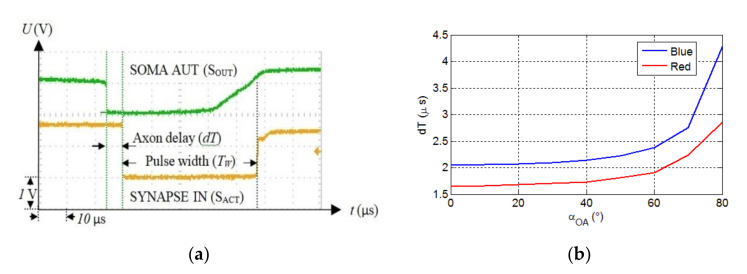
(**a**) The measured time signals showing the soma output (upper signal) and the synapse input (lower signal), which is delayed by *dT*, and (**b**) the axon delay vs. $\alpha_{\mathrm{OA}}$ for the red and blue LEDs and for $L_{\mathrm{OA}}\;=\;50\;\mathrm{cm}.$

**Figure 7 sensors-20-06119-f007:**
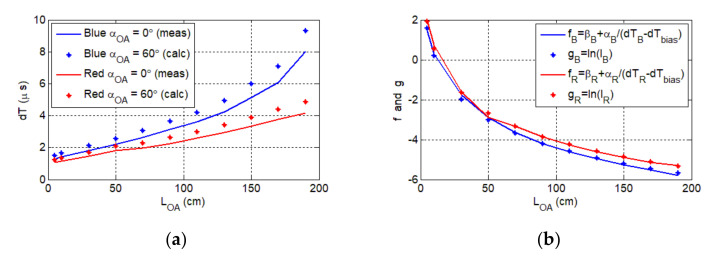
(**a**) The delay dT vs. the optical link length $L_{\mathrm{OA}}$ (full line) for the aligned blue and red channels. The calculated axon delay for the maximum misalignment angle $\alpha_{\mathrm{OA}}=60{}^{\circ}$ (dotted line). (**b**) The graphical representation of the relation between the photocurrent I and the delay dT, respectively, for the red channel (red line and dots) and the blue channel (blue line and dots).

**Figure 8 sensors-20-06119-f008:**
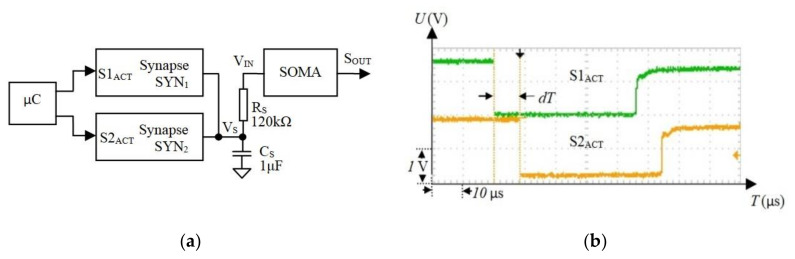
(**a**) Structure of the spiking neural network (SNN), which includes two synapses and a soma for testing the influence of the axon delay on the activations of the neurons; (**b**) the microcontroller was programmed to generate delays dT between pulses that activate two synapses through inputs $S1_{\mathrm{ACT}}$ and $S2_{\mathrm{ACT}}$.

**Table 1 sensors-20-06119-t001:** System parameters.

Parameter	Value
LEDs (CREE XLamp ML-E)	
View angle	125°
Red—Wavelength, drive current,	640 nm, 178 mA,
$\mathrm{Illuminance}\;E_{V}^{\mathrm{init}}\;\mathrm{for}\;10\;\mathrm{cm}\;\mathrm{channel}\;\mathrm{length}$	460 lux
Blue—Wavelength, drive current,	465 nm, 146 mA,
$\mathrm{Illuminance}\;E_{V}^{\mathrm{init}}\;\mathrm{for}\;10\;\mathrm{cm}\;\mathrm{channel}\;\mathrm{length}$	252 lux
Optical channel length for $E_{V}^{\mathrm{init}}$	10 cm
Optical filter (Edmund Optics)	
Red—Centre λ, FWHM, Transmittance	634 nm, 71 nm, 93%
Blue—Centre λ, FWHM, Transmittance	470 nm, 85 nm, 95%
ORx (Thorlabs PDA100A2)	
Gain (HiZ)	4.75 × 10^6^ V/A ±5%
Bandwidth	3 kHz
Responsivity (A/W)	0.25 (B) and 0.45 (R)
Sensitivity	340–1100 nm
V_DD_ and V_CC_	1.6 and 4.5 V
Room illuminance from the artificial source	440–462 lux.

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
