# Peer review of "Optical Axons for Electro-Optical Neural Networks"

_sensors, 2020, doi:10.3390/s20216119_

Round 1

Reviewer 1 Report

This paper presents the design and experimental validation of an artificial neural network that uses optical axons instead of optical synapses to eliminate the need to transmit weighted optical signals.

The paper is well written, the methods and background are clearly presented and the results are convincing, thus it can be accepted for publication as it is. 

Author Response

We would like to deeply thank the respected reviewer for the time and efforts he/she spent in reviewing our paper.

Reviewer 2 Report

This study proposed optical axons for electro-optical spiking neural networks. The topic is interesting and can contribute for the development of mobile neuromorphic sensors.

The following recommendations are provided for the authors' reference.

  1. [Abstract] Please provide some quantitative findings in the abstract.
  2. The discussion section is too short. It would be helpful, if authors more elaborate this section.
  3. How will authors implement this approach in practice?
  4. What is next step? What are the current limitations and what needs to be done to make this approach more robust and generic?

Author Response

We would like to deeply thank the reviewer for the time and efforts. Also, we highly appreciate your supporting comments. In the attachment, we have tried our best to address all the reviewer comments and concerns in the revised manuscript.

Reviewer 3 Report

Letter  Optical Axons for Electro-Optical Neural Networks

Sensors

  • In abstract, in lines 13 to 24 is many introductioin, and of 25 to 29 is the objective, methodology and resulted. There are more in introduction that scientific contribution. Also, it is necessary to add conclusion in the abstract.
  • Clarify, in the section experimental setup. As the experiments are developed?, the authors speak of different parameters evaluated at different intervals, in this case, it is necessary how many experiments were carried out and how they were carried out?.
  • A nomenclature is necessary.

Author Response

We would like to thank the reviewer for their constructive comments. They have been very helpful in the revision of this paper and allowed us to improve the technical contents and presentation quality.

Reviewer 4 Report

This paper presents an Optical Axon (OA) system for Optical Neural Networks. Specifically, the paper contributes to the general field of Neuromorphic engineering which aims at mimicking biological Neural Networks using electronic hardware. A recent subfield is represented by optical neural networks where the information transmission between neurons is achieved through pulsed signals in the optical spectrum. The paper follows a good methodology and it is generally well written. There are three main issues, and some minor, this reviewer would like to be clarified.

  • The first main question regards the motivation for this work which is not entirely clear. The authors motivate the use of OAs only in lines 96-98. Although this statement is reasonable, the paper does not contain any tangible elements that the mentioned problems were relevant disadvantages for ONNs in the first place. It would be required to characterize those problems and show how the OA solves or improves on  them.

  • The second main question is about the scalability of the system. Since parallelism is one of the claims raised in the introduction, it was not obvious for this reviewer (and possibly for some other readers) how the OAs scale to multiple synapses in parallel. What is the fan-in and fan-out of the system?

  • The third main point touches upon the general practice of scientific contributions to have repeated measurements and to report average and standard deviations of those. All figures and reported numbers seem the result of a single measurement. If there is a specific reason for that, it should be clarified in the text.

Finally, some general questions and minor observations:

  • This reviewer would also be interested in understanding if the transmission delay could be influenced by the firing frequency of the SNNs and thus if mechanisms such as firing rate adaptation [1-5 not exhaustive list] could be implemented in your system.
  • In the introduction when listing the benefits of SNNs, it should be mentioned first the energy consumption reduction when compared to ANNs [6-8 not exhaustive list].
  • A famous paper from Izhikevich in 2006 proposed a learning method based on axon delays; this reviewer found the link with the presented paper quite interesting.
  • “if” in eq. 3 and 4 have different font.

[1] Fairhall, A. L., Lewen, G. D., Bialek, W., & van Steveninck, R. R. D. R. (2001). Efficiency and ambiguity in an adaptive neural code. Nature412(6849), 787-792.

[2] Pozzorini, C., Naud, R., Mensi, S., & Gerstner, W. (2013). Temporal whitening by power-law adaptation in neocortical neurons. Nature neuroscience, 16(7), 942-948.

Chicago

[3] Jolivet, R., Rauch, A., Lüscher, H. R., & Gerstner, W. (2006). Predicting spike timing of neocortical pyramidal neurons by simple threshold models. Journal of computational neuroscience21(1), 35-49.

[4] Bohte, S. M. (2012). Efficient spike-coding with multiplicative adaptation in a spike response model. In Advances in Neural Information Processing Systems (pp. 1835-1843).

[5] Zambrano, D., Nusselder, R., Scholte, H. S., & Bohté, S. M. (2019). Sparse computation in adaptive spiking neural networks. Frontiers in neuroscience, 12, 987.

[6] Liu, S. C., & Delbruck, T. (2010). Neuromorphic sensory systems. Current opinion in neurobiology, 20(3), 288-295.

[7] Boahen, K. (2017). A neuromorph's prospectus. Computing in Science & Engineering19(2), 14-28.

[8] Esser, S. K., Merolla, P. A., Arthur, J. V., Cassidy, A. S., Appuswamy, R., Andreopoulos, A., ... & di Nolfo, C. (2016). From the cover: convolutional networks for fast, energy-efficient neuromorphic computing. Proceedings of the National Academy of Sciences of the United States of America, 113(41), 11441. Chicago           

[9] Izhikevich, E. M. (2006). Polychronization: computation with spikes. Neural computation18(2), 245-282.

Author Response

Thank you for your supporting comments and the points raised. They have been very helpful in the revision of this paper and allowed us to improve the technical contents and presentation quality. We have addressed all comments, see attachment, and revised the paper accordingly.

Round 2

Reviewer 3 Report

Thank you